# A Comparison of the Meat Quality, Nutritional Composition, Carcass Traits, and Fiber Characteristics of Different Muscular Tissues between Aged Indigenous Chickens and Commercial Laying Hens

**DOI:** 10.3390/foods12193680

**Published:** 2023-10-07

**Authors:** Li Liu, Qian Chen, Lingqian Yin, Yuan Tang, Zhongzhen Lin, Donghao Zhang, Yiping Liu

**Affiliations:** Farm Animal Genetic Resources Exploration and Innovation Key Laboratory of Sichuan Province, Sichuan Agricultural University, Chengdu 611130, China

**Keywords:** chicken, breast muscle, thigh muscle, meat quality, nutrition, amino acids

## Abstract

The aim of this study is to assess the differences in the meat quality, nutritional composition, carcass traits, and myofiber characteristics between Hy-Line grey chickens (HLG, commercial breed) and Guangyuan grey chickens (GYG, indigenous breed). A total of 20 55-week-old chickens were selected for slaughter. The HLG exhibited a larger carcass weight, breast muscle weight, and abdominal fat weight (*p* < 0.05). The GYG exhibited a higher crude protein content, lower shear force, and smaller fiber size in the thigh muscles, whereas the HLG presented higher pH values and lower inosine-5′-monophosphate content in the breast muscles (*p* < 0.05). Darker meat based on higher redness and yellowness values was observed in the GYG instead of the HLG (*p* < 0.05). The research results also revealed parameter differences between different muscle types. Simultaneously, a correlation analysis showed significant correlations between the meat quality traits and myofiber characteristics (*p* < 0.05). In conclusion, aged indigenous chickens perform better in terms of tenderness and nutritional value in the thigh muscles, and may exhibit a better flavor in the breast muscles, but have a smaller breast muscle weight. Therefore, the current investigation provides a theoretical basis for the different needs of consumers and the processing of meat from old laying hens.

## 1. Introduction

Chicken meat has become widely appreciated by consumers due to its healthy nutritional components of high protein, low fat, and low cholesterol [1]. In addition, chicken products are not only affordable, but also have no cultural or religious restrictions [2,3], which has further promoted the expansion of the chicken market. In the market, in addition to broilers specifically developed for meat production, laying hens are also an important source of poultry meat, particularly old laying hens that need to be utilized.

In the late stage of egg laying (from week 80 to 100), laying hens will gradually be removed from farms due to the decline in their egg production rate or egg quality; these hens are named “spent hens” and are also classified as old laying hens or aged chickens [4,5]. It is estimated that 4.5 billion spent hens were produced globally in 2018 [6]. Undoubtedly, as the largest egg-producing country [7], China has a large number of old laying hens that are eliminated every year, including the main commercial egg-producing breed, Hy-Line gray layers [8]. Therefore, how to properly handle these old laying hens has become an urgent priority. At present, from a commercial point of view, processing the meat of old laying hens into diversified value-added products is an important way of obtaining residual value in the egg production industry and represents a wise choice from the perspective of environmental protection [4]. The meat of old laying hens is generally tough and chewy, which has a negative impact on its cooking quality and technological properties. Nonetheless, in some parts of the world, particularly in China, aged chickens are still processed into many popular table delicacies, such as seek kababs, chicken nuggets, and chicken broth [9,10,11].

Guangyuan grey chicken is an indigenous chicken breed in China [12], known for its typical phenotypic characteristics, namely gray feathers and skin [10]. Compared to commercial breeds, most consumers have a stronger willingness to pay for aged indigenous chicken because they believe it will perform better in terms of nutrition and flavor [13,14]; however, this has led to higher market prices for indigenous chickens. Currently, numerous studies have reported differences in the meat quality and nutritional composition between commercial broiler chickens and indigenous chickens. However, there is a scarcity of information regarding the differences between commercial laying hens and indigenous chickens, especially spent chickens. Thus, we compared the meat quality, nutritional composition, carcass traits, and myofiber characteristics of the breast and thigh muscles between HLG and GYG. The purpose of this study is to present the differences in the meat between old laying hens and indigenous chickens for consumers, further providing more possibilities for consumers to choose and providing a theoretical basis for the effective utilization of meat from old laying hens.

## 2. Material and Methods

### 2.1. Animals and Sample Collection

All the birds used in this experiment were raised in Sichuan Tianguan Ecological Agriculture and Husbandry Co., Ltd. (Guangyuan, China). A total of 120 one-day-old female chickens were reared up to 55 weeks of age, including Hy-Line grey chickens (HLG, *n* = 60) and Guangyuan grey chickens (GYG, *n* = 60). All of the experimental chickens were raised under the same conditions, without any significant differences in nutrition and environment, and had free access to food and water. The dietary composition and nutritional content at all stages are shown in Appendix A. After 12 h of fasting, ten healthy chickens were randomly selected from each breed for this trial. The chickens were sacrificed by cutting their carotid arteries after breathing anesthesia and then deplumation. The carcass weight, eviscerated yield weight, and semi-eviscerated weight were determined using the method described by Yuan et al. [15]. The breast muscle, thigh muscle, and abdominal fat were carefully dissected from the carcasses and weighed, and their percentages were calculated, respectively. The right sides of the breast and thigh muscles were used to evaluate the meat quality parameters within 24 h, including the meat color, pH, drip loss, cooking loss, and shear force. The remaining half of meat was used for the chemical composition, myofiber characteristics, and amino acid profile analyses. All the experimental procedures were conducted in accordance with the principles and procedures approved by Sichuan Agricultural University’s Animal Care and Use Committee (approval no. DKY2021202052).

### 2.2. pH and Meat Color

A portable pH meter (TEST0205, Shanghai, China) was used to measure the pH value at 24 h postmortem. The color parameters of each sample were evaluated using a colorimeter (CR-300, Konica Minolta, Japan), using CIE L* (lightness), CIE a* (redness), and CIE b* (yellowness) values. The pH and color values were taken from different positions of the meat, and the final results are represented as an average of three repetitions. 

### 2.3. Drip Loss, Cooking Loss, and Shear Force

The drip loss was measured by the weight difference between the suspended meat before and after low-temperature storage. Approximately 5 g of chicken meat, with the shape of being roughly parallelepiped, was hung in a sealable plastic bag and placed at 4 °C for 48 h. The parallelepiped meat was reweighed after removing the excess water using absorbent papers carefully.

The cooking loss of the samples was determined by their weight loss after cooking. In more detail, the raw meat was weighed and removed anadesma, then packaged and sealed in a polyethylene bag, and subsequently cooked in a water bath at 85 °C for 10 min. The heat-treated samples were cooled for 20 min at room temperature (25 °C), and subsequently reweighed after removing the excess water using absorbent papers. The results of the drip and cooking losses are presented as percentages of the original sample weight. 

The shear force was determined using a C-TM3 digital display meat tenderness meter (Northeast Agricultural University, Harbin, China). The cooled samples used for the cooking loss were cut into strips (4.5 × 1.5 × 1.5 cm each) parallel to the muscle fiber orientation to measure the shear force value. Each strip was measured perpendicularly to the direction of the muscle fibers in three replicates, and the average value was calculated.

### 2.4. Chemical Composition

According to the National Standard of China (GB 5009.5-2016, 5009.6-2016), the breast and thigh muscles of each breed were employed to assess the contents of crude protein (CP) and crude fat (CF). The moisture content was calculated from the difference in the weight of the meat before and after 24 h of drying at 105 °C, and the percentage was generated.

A weight of 5 g of crushed fresh muscle samples was used for an inosine-5′-monophosphate (IMP) analysis. The sample was placed in a 50 mL centrifuge tube and 10 mL of 6% perchloric acid was added carefully. An equal amount of perchloric acid was added again, and the above steps were repeated to collect the homogenate. The homogenate was centrifuged at 8000 rpm for 13 min, and the supernatant was filtered into a beaker. After adjusting the pH value to 6.5 using 0.5 and 5 mol/L NaOH solutions, the solution was transferred to a 100 mL volumetric flask to volume, and then filtered with 0.25 µm for a high-performance liquid chromatography analysis.

### 2.5. Amino Acid Profile

Based on the described method [10], the content and composition of the amino acids were measured using an automatic amino acid analyzer (S433D, Sykam, Munich, Germany) after drying and smashing the meat samples. All the procedures for analyzing the amino acid content were carried out in accordance with the regulations and requirements of the Chinese national standard GB 5009.124-2016.

### 2.6. Myofiber Characteristics

The strips (2 × 1 × 1 cm, each) from the same position of the pectoralis major (*m. pectoralis major*) and thigh muscle (*m. biceps femoris*) were cut with a surgical blade perpendicular to the direction of the muscle fiber. The meat samples were fixed in a 4% paraformaldehyde solution for 48 h, and then stained with hematoxylin and eosin for further observation. Micrographs were taken using a digital microscope (BA400Digital, Xiamen, China) with a magnification of 100 times. The obtained images of each sample were used to determine the diameter, cross-sectional area (CSA), and density of the muscle fibers using Image-Pro Plus Version 6.0 software (Media Cybernetics, Inc., Rockville, MD, USA). The CSA of the breast and thigh muscles was determined by selecting 30 muscle fibers that were intact and well-orientated. Their diameter and density were calculated using the method described by Liu et al. [16]. The fields on each slice were randomly selected, and the average value was used for the data analysis. 

### 2.7. Statistical Analysis

The data were analyzed using a one-way analysis of variance (ANOVA) implemented in SPSS version 27.0 (SPSS, Inc., Chicago, IL, USA). The independent-samples t-test was used for a significant difference analysis between the different groups. A Pearson’s correlation analysis was performed to assess the correlation of the muscle fiber characteristics and the meat quality traits. The obtained data are expressed as the mean ± standard error of the mean (SEM). A value of *p* < 0.05 was considered to be statistically significant.

## 3. Results

### 3.1. Carcass Traits

The results of the carcass traits of the HLG and GYG are depicted in Table 1. Compared to the GYG, the carcass weight, breast muscle weight, abdominal fat weight, and abdominal fat yield of the HLG were heavier (*p* < 0.05). However, the thigh muscle yield of the GYG was larger than that of the HLG (*p* < 0.05).

### 3.2. pH and Meat Color

The pH values and meat color of the breast and thigh muscles from the HLG and GYG are presented in Table 2. The pH value of the thigh muscle was significantly higher than that of the breast muscle, regardless of the chicken breed (*p* < 0.05). Compared to the GYG, the breast muscle of the HLG showed higher pH values (*p* < 0.05). Within the same breed, the thigh muscle exhibited a higher pH value (*p* < 0.05).

The meat colors of the breast and thigh muscles from the HLG and GYG were also evaluated. For the two breeds, the L* value in the breast muscle was remarkably higher than that in the thigh muscle (*p* < 0.05). In addition, there was no statistical significance in the L* values between the two breeds. The a* value of thigh muscle was significantly higher than that of breast muscle, regardless of the chicken breeds(*p* < 0.05). There were no significant differences in the a* values of the breast and thigh muscles between the two breeds. Considering the b* value, the b* value of the thigh muscle was higher than that of the breast muscle, regardless of the chicken breed (*p* < 0.05). The thigh muscle of the GYG showed higher b* values than that of the HLG (*p* < 0.05).

### 3.3. Drip Loss, Cooking Loss, and Shear Force

The shear force, drip, and cooking losses of the breast and thigh muscles from the HLG and GYG are illustrated in Table 3. Regardless of the chicken breed, the cooking loss of the thigh muscle was significantly higher than that of the breast muscle (*p* < 0.05), while the drip loss showed a reverse trend and did not reach a significant level (*p* > 0.05). Furthermore, no appreciable differences were observed in the drip and cooking losses between these two breeds. Regardless of the chicken breed, the shear force value in the thigh muscle was higher than that in the breast muscle. Compared to the HLG, the GYG presented higher shear force values in the thigh muscles, but there was no significant difference between the breast muscles of different breeds.

### 3.4. Meat Chemical Analysis

The chemical compositions of the breast and thigh muscles from the HLG and GYG are shown in Figure 1. The results showed that the CP content in the breast muscle was significantly higher (*p* < 0.05) than that in the thigh muscle, regardless of the chicken breed. Regardless of the two breeds, the thigh muscle showed higher CF and moisture contents as compared to the breast muscle (*p* < 0.05). For different breeds, the CP content in the thigh muscles of the GYG was higher than that of the HLG (*p* > 0.05). Furthermore, no effect of chicken breed on the CF and moisture content was observed in the current research. Likewise, the IMP content in the breast muscles of the HLG was lower than that in the other groups.

### 3.5. Amino Acid Profile

The amino acid contents of the breast and thigh muscles from the different chicken breeds are depicted in Table 4. On the whole, there were no statistically significant differences in the amino acid contents between the different breeds in their breast and thigh muscles, with the exception of His, Ser, Glu, Cys, and EAA/NEAA. Specifically, regardless of the chicken breed, the E/NE in the breast muscle was higher (*p* < 0.05) than that in the thigh muscle, and the content of His was also the same. For these two breeds, the contents of Ser, Glu, and Cys in the thigh muscle were higher than those in the breast muscle. Additionally, it was found that the content of Lys in the essential fraction was the highest among all the groups, followed by Leu and Arg. Considering the non-essential fraction, Glu was the most abundant in content, followed by Asp, Ala, and Gly, while the content of Cys was the lowest amino acid in all the samples. 

### 3.6. Myofiber Characteristics

A comparison of the myofiber characteristics between the different groups is indicated in Figure 2. The results showed that the myofiber characteristics of the breast muscle were not affected by the breeds. For the thigh muscles, the HLG had a larger myofiber diameter and cross-sectional area when compared to the GYG (*p* < 0.05), while their muscle fiber density was smaller. For the different muscle types, the myofiber diameter and CSA in the thigh muscles were larger than those in the breast muscles, while the myofiber density showed a reverse trend.

### 3.7. Correlations

Regarding the correlation between the meat quality traits, the results showed that the pH was significantly negatively correlated with the drip loss, as shown in Figure 3. Moreover, the pH had a moderate correlation with L*, a*, and b*, but did not reach a significant level. L* was positively correlated with the drip loss, and negatively correlated with a*, the cooking loss, and the crude fat. In contrast, a* was positively correlated with the cooking loss (*p* < 0.01) and crude fat (*p* < 0.05). The cooking loss was significantly positively correlated with the crude fat. The shear force was significantly negatively correlated with the crude protein, while positively correlated with the crude fat. Regarding the correlation between the myofiber characteristics and the meat quality traits, our results showed that there was a moderate or significant correlation between the fiber characteristics and the pH, L*, a*, cooking loss, drip loss, shear force, crude protein, and crude fat, as well as the moisture content in both the GYG and HLG. In addition, the myofiber diameter was significantly positively correlated with the CSA and negatively correlated with the myofiber density.

## 4. Discussion

Generally, slow-growing indigenous breeds have less abdominal fat [17]. In the current study, the HLG had heavier abdominal fat compared to the GYG. Furthermore, the HLG had a greater carcass weight and breast muscle weight. These may have been related to differences in the metabolic rates between the different breeds. According to reports, commercial chickens that have been intensively selected are more likely to perform prominently in terms of breast muscle yield and abdominal fat weight compared to unselected lines [18,19]. Particularly for commercial laying hens, such as HLG, they are more likely to exhibit an excessive fat accumulation in late-phase hens due to vigorous lipid metabolism and oxidation [20]. However, this physiological condition has a negative impact on egg production efficiency, further accelerating the process of being eliminated [21]. 

pH is considered to be an essential indicator for evaluating the quality of poultry meat, as it may directly affect the texture, color, and flavor of the meat [22]. As expected, the pH value in the thigh muscle was higher than that in the breast muscle, regardless of the chicken breed, which is in agreement with the previous study [23]. These findings can be explained by the metabolic differences caused by the composition of the fiber types in the meat. To our knowledge, after animals are slaughtered, glycogen is broken down into glucose, and then glucose undergoes glycolysis to form lactic acid, leading to a decrease in pH value [24]. However, there are more glycolytic fibers in breast muscles than in thigh muscles [25], suggesting that there is a higher content of glycogen in thigh muscle for glycolysis [26]. Regarding the differences between the breeds, the pH values in the breast and thigh muscles of the HLG were higher than those of the GYG, whereas significant differences were only observed between the breast muscles of the two breeds. This phenomenon may have been caused by the heavier carcass weight, as the chickens with heavier carcass weights tended to exhibit higher pre-slaughter stress, leading to a decrease in the glycogen in the meat, especially in older chickens [27,28]. 

The color of meat is regarded as one of the important meat quality parameters, since it may have a direct impact on consumers’ purchasing desires [29,30,31]. According to reports [32], raw breast meat appears as pale pink, while raw thigh meat presents as dark red. In the present investigation, we observed that the a* and b* values in the thigh muscles were remarkably higher than those in the breast muscles, regardless of the chicken breed, but this was reversed for the L* values, which was consistent with their apparent characteristics. Furthermore, the results showed that the L*, a*, and b* values of the GYG were higher than those of the HLG, and significant differences were observed in the b* values between the thigh muscles, implying that GYG have a darker meat color than HLG. This is similar to previous reports which have demonstrated that indigenous breeds have a darker meat than improved breeds [33,34]. Simultaneously, it was reported that differences in meat color are also associated with the concentration differences of pigmentation, myoglobin, and hemoglobin [15,35]. 

Water-holding capacity has a significant impact on the tenderness and juiciness of meat products [36], which is determined based on the dripping loss and cooking loss. When heated, the protein in the meat is degraded, causing the breast meat that contains more protein to release more juice [37]. This may be the reason why the cooking loss in the breast muscle was significantly higher than that in the thigh muscle; however, we did not obtain similar findings in this trial. The previous study demonstrated that a high fat content is associated with a high water-binding capacity [38]. In the current study, the thigh muscles with a higher fat content had a lower water-holding capacity in the cooked meat and a higher water-holding capacity in the raw meat, respectively, manifested as a higher cooking loss and lower dripping loss. Concerning the shear force, the shear force values in the thigh muscles were higher than those in the breast muscles, as previously reported [39]. Furthermore, compared to the GYG, the HLG presented higher shear force values in the thigh muscles, indicating that the meat of commercial laying hens is chewier than that of indigenous local broilers. 

Our results concur with the previous findings which revealed that muscle type can affect the contents of CF, CP, and moisture [40]. Specifically, regardless of the chicken breed, the CF content and moisture were significantly higher in the thigh muscle as compared to the breast muscle, whereas this was reversed for the CP content. It was reported that the thigh muscle had more oxidative fibers than the breast muscle [41]. Moreover, the intramuscular fat content of oxidized meat is higher compared to breast muscle that contains more glycolytic muscle fibers [42]. Our results can be explained by these findings. For the different breeds, no significant differences were observed in the CF and moisture contents. However, the results showed that the CP content in the breast muscles of the GYG and HLG was significantly higher than that in the thigh muscles. Likewise, the IMP content in the breast muscle of the HLG was lower than that of the other groups. As a result, we believe that aged indigenous broilers may exhibit more advantages in terms of meat flavor and nutrition as compared to commercial laying hens. Interestingly, concerning the amino acid profile, there was no statistically significant difference between the breeds and muscle types. This may have been due to the fact that commercial laying hens and aged indigenous broilers are selected under the same feeding management.

Myofiber characteristics have been well-demonstrated to be key contributing factors to meat quality and quantity [41,43]. In this study, regardless of the two breeds, the myofiber diameter and CSA in the thigh muscles were larger than those in the breast muscles, while the myofiber density showed a reverse trend, which is consistent with the previous study on Pekin ducks [44]. For the different muscle types, the results showed that there was no difference in the muscle fiber characteristics of the breast muscles between the different breeds. The thigh muscle of the HLG had a larger muscle fiber diameter and CSA as compared to the GYG, and the muscle fiber density was smaller. These results correspond to higher shear forces, which may lead to tougher thigh meat. 

In the current study, we also evaluated the correlation between the meat quality traits of the two breeds. Numerous studies have suggested that poultry meat pH is highly correlated with meat color [45,46,47]. These findings were confirmed in this study, summarizing that the pH was negatively correlated with L* and positively correlated with a* or b*, but did not reach a significant level. In addition, there was a moderate or significant correlation between water-holding capacity and pH or meat color in this study. Previous studies have confirmed that higher pH values are closely related to a lower drip loss and higher cooking loss, respectively [48]. It has elsewhere been reported that water-holding capacity has a high correlation with the L* value of breast muscle [49]. This correlation analysis is consistent with our results. The significance of myofiber characteristics towards poultry meat quality has been extensively debated in recent years. The present study demonstrated that there was a moderate or significant correlation between the muscle fiber characteristics and meat quality parameters, indicating that muscle fiber characteristics have an impact on meat quality. A recent study confirmed that the muscle fiber characteristics were significantly correlated with the meat pH, shear stress, and protein content in slow-growing and fast-growing ducks, and there are differences in the correlations with other characteristics between breeds [50]. Thus, we speculate that these differences are influenced by breed, age, and muscle type, which require further attention in future research.

## 5. Conclusions

In this study, we evaluated the differences in the meat quality, nutritional composition, carcass traits, and myofiber characteristics between aged indigenous broilers and commercial laying hens. The aged indigenous chickens performed better in terms of tenderness and nutritional value in the thigh muscles, and may exhibit a better flavor in the breast muscles, but had a smaller breast muscle weight. There were differences in the meat quality parameters, nutritional properties, and myofiber characteristics between the breast and thigh muscles, regardless of breed. The correlation coefficients obtained in this study demonstrated that myofiber characteristics may play an important role in meat quality parameters.

## Figures and Tables

**Figure 1 foods-12-03680-f001:**
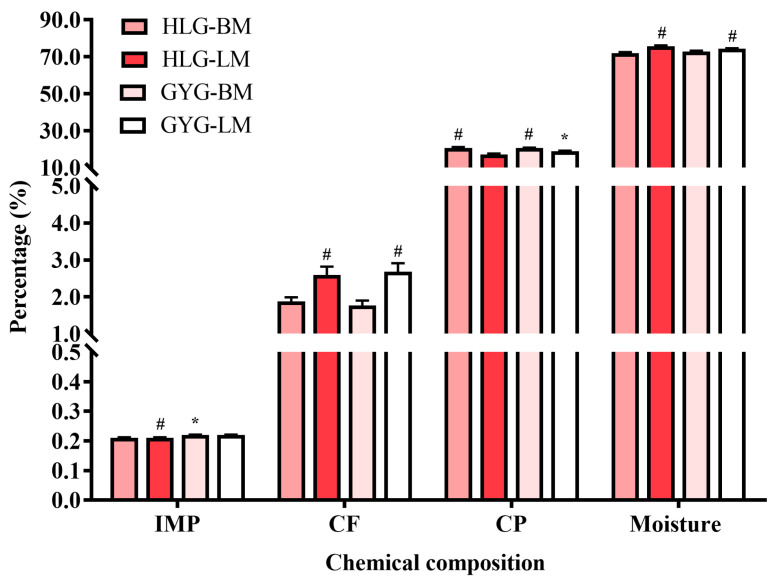
Chemical composition analysis of breast and thigh muscles (g/100 g wet weight) from HLG and GYG (*n* = 10/breed). ^*^ Superscripted in means indicates significant differences between breeds (*p* < 0.05). ^#^ superscripted in means indicates significant differences between muscle parts within same breeds (*p* < 0.05). HLG-BM, the breast muscle of Hy-Line grey chickens; HLG-TM, the thigh muscle of Hy-Line grey chickens; GYG-BM, the breast muscle of Guangyuan grey chickens; GYG-TM, the thigh muscle of Guangyuan grey chickens; IMP, inosine-5′-monophosphate; CF, crude fat; and CP, crude protein.

**Figure 2 foods-12-03680-f002:**
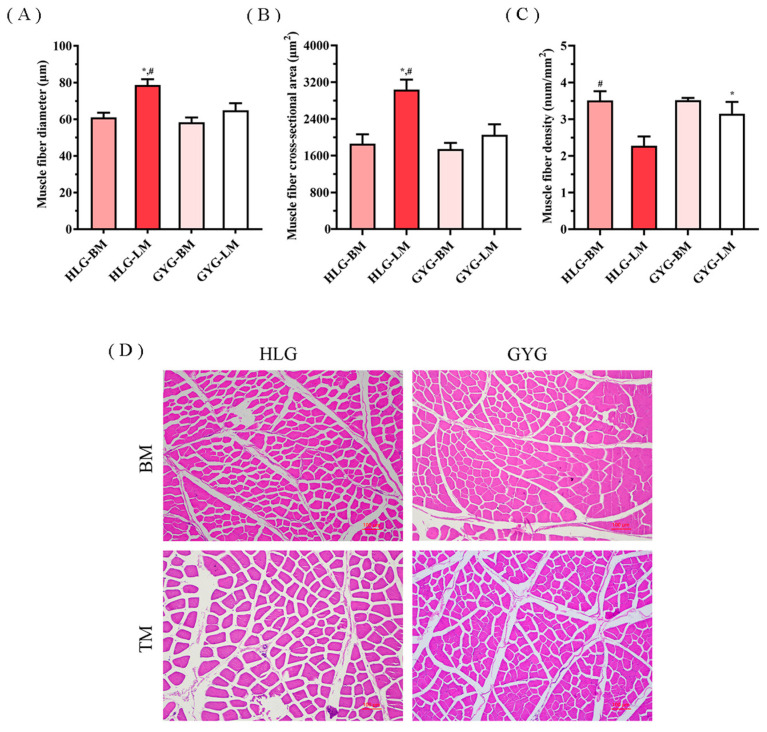
Myofiber characteristics evaluation of breast and thigh muscles from HLG and GYG (*n* = 3/breed). (**A**) diameter; (**B**) cross-sectional area; (**C**) density; and (**D**) histologic characteristics. All slices were prepared under the same conditions (H&E, 100×), and the bars were 100 μm. ^*^ Superscripted in means indicates significant differences between breeds (*p* < 0.05). ^#^ superscripted in means indicates significant differences between muscle parts within same breeds (*p* < 0.05). HLG-BM, the breast muscle of Hy-Line grey chickens; HLG-TM, the thigh muscle of Hy-Line grey chickens; GYG-BM, the breast muscle of Guangyuan grey chickens; GYG-TM, the thigh muscle of Guangyuan grey chickens; BM, breast muscle; and TM, thigh muscle.

**Figure 3 foods-12-03680-f003:**
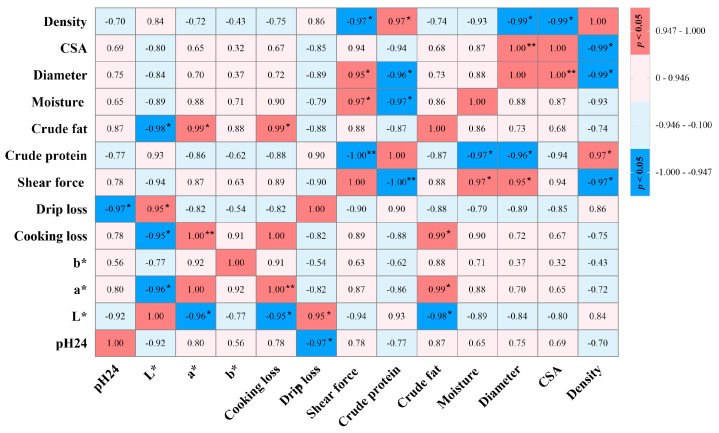
Heat map of correlation coefficients (r) between myofiber characteristics and meat quality traits of breast and thigh muscles in HLG and GYG. Significance is denoted by superscripts * and **, at *p* < 0.05 and *p* < 0.01, respectively. CSA, myofiber cross-sectional area.

**Table 1 foods-12-03680-t001:** Carcass traits from HLG and GYG (*n* = 10/breed).

Traits	HLG	GYG	*p*-Value
Carcass weight (g)	1583.00 ± 62.31	1345.00 ± 60.80	*
Eviscerated weight (g)	1174.00 ± 52.39	1069.00 ± 31.64	NS
Semi-eviscerated weight (g)	1039.40 ± 45.90	920.00 ± 39.75	NS
Breast muscle weight (g)	175.58 ± 8.21	135.55 ± 9.88	*
Breast muscle yield (%)	16.95 ± 0.50	14.80 ± 0.95	NS
Thigh muscle weight (g)	229.26 ± 11.29	221.67 ± 7.97	NS
Thigh muscle yield (%)	22.03 ± 0.38	24.31 ± 0.88	*
Abdominal fat weight (g)	51.26 ± 6.49	21.88 ± 3.17	*
Abdominal fat yield (%)	4.54 ± 0.43	2.37 ± 0.36	*

An asterisk (*) denotes a significant difference in different breeds (*p* < 0.05). NS, nonsignificance.

**Table 2 foods-12-03680-t002:** pH and meat color of breast and thigh muscles from HLG and GYG (*n* = 10/breed).

Traits	HLG-BM	HLG-TM	GYG-BM	GYG-TM
pH	6.00 ± 0.04 ^*^	6.16 ± 0.02 ^#^	5.74 ± 0.01	6.13 ± 0.02 ^#^
L*	40.16 ± 0.53 ^#^	35.50 ± 0.85	41.44 ± 0.50 ^#^	36.23 ± 0.82
a*	2.28 ± 0.10	9.92 ± 0.60 ^#^	2.60 ± 0.16	11.14 ± 0.70 ^#^
b*	8.02 ± 0.25	9.58 ± 0.50 ^#^	8.73 ± 0.26	10.88 ± 0.45 ^*,#^

^*^ Superscripted in means indicates significant differences between breeds (*p* < 0.05). ^#^ superscripted in means indicates significant differences between muscle parts within same breeds (*p* < 0.05). HLG-BM, the breast muscle of Hy-Line grey chickens; HLG-TM, the thigh muscle of Hy-Line grey chickens; GYG-BM, the breast muscle of Guangyuan grey chickens; GYG-TM, the thigh muscle of Guangyuan grey chickens; L*, lightness; a*, redness; and b*, yellowness.

**Table 3 foods-12-03680-t003:** Drip loss, cooking loss, and shear force of breast and thigh muscles from HLG and GYG (*n* = 10/breed).

Traits	HLG-BM	HLG-TM	GYG-BM	GYG-TM
Cooking loss (%)	25.82 ± 1.03	35.10 ± 0.80 ^#^	26.72 ± 0.92	35.99 ± 0.74 ^#^
Drip loss (%)	2.70 ± 0.23	2.58 ± 0.17	2.79 ± 0.20	2.64 ± 0.19
Shear force (kg)	3.45 ± 0.23	5.86 ± 0.14 ^*,#^	3.48 ± 0.21	4.75 ± 0.17 ^#^

^*^ Superscripted in means indicates significant differences between breeds (*p* < 0.05). ^#^ superscripted in means indicates significant differences between muscle parts within same breeds (*p* < 0.05). HLG-BM, the breast muscle of Hy-Line grey chickens; HLG-TM, the thigh muscle of Hy-Line grey chickens; GYG-BM, the breast muscle of Guangyuan grey chickens; and GYG-TM, the thigh muscle of Guangyuan grey chickens.

**Table 4 foods-12-03680-t004:** Amino acid contents of breast and thigh muscles (g/100 g dry weight) from HLG and GYG (*n* = 10/breed).

	Amino Acid	HLG-BM	HLG-TM	GYG-BM	GYG-TM
	Thr ^▲^	3.73 ± 0.01	3.74 ± 0.12	3.62 ± 0.09	3.94 ± 0.02
	Val ^△^	3.83 ± 0.08	3.65 ± 0.22	3.72 ± 0.10	3.83 ± 0.05
	Met ^△^	2.40 ± 0.06	2.45 ± 0.15	2.32 ± 0.06	2.58 ± 0.04
	Ile ^△^	3.79 ± 0.09	3.66 ± 0.21	3.64 ± 0.12	3.85 ± 0.04
Essential	Leu ^△^	6.60 ± 0.16	6.29 ± 0.30	6.44 ± 0.15	6.63 ± 0.07
	Phe ^△^	3.26 ± 0.08	3.11 ± 0.15	3.21 ± 0.06	3.29 ± 0.03
	Lys	7.41 ± 0.17	7.30 ± 0.45	7.17 ± 0.21	7.66 ± 0.10
	His	3.53 ± 0.06 ^*,#^	2.44 ± 0.23	2.93 ± 0.18 ^#^	2.33 ± 0.04
	Arg ^△^	5.37 ± 0.09	5.40 ± 0.27	5.26 ± 0.11	5.68 ± 0.08
	EAA	39.91 ± 0.86	38.04 ± 2.16	38.32 ± 1.07	39.79 ± 0.45
	Asp ^▲^	7.71 ± 0.17	7.59 ± 0.40	7.53 ± 0.18	8.01 ± 0.09
	Ser ^▲^	3.23 ± 0.09	3.44 ± 0.15	3.18 ± 0.06	3.67 ± 0.02 ^#^
	Glu ^▲^	12.37 ± 0.29	13.13 ± 0.61	12.00 ± 0.31	13.77 ± 0.19 ^#^
	Gly ^▲^	3.31 ± 0.09	3.47 ± 0.17	3.43 ± 0.10	3.73 ± 0.10
	Ala ^▲^	4.66 ± 0.11	4.51 ± 0.25	4.58 ± 0.08	4.77 ± 0.04
Non-essential	Cys	0.40 ± 0.02	0.51 ± 0.03 ^#^	0.39 ± 0.01	0.52 ± 0.00 ^#^
	Tyr	2.90 ± 0.05	2.89 ± 0.10	2.85 ± 0.06	3.04 ± 0.03
	Pro ^△^	2.82 ± 0.06	2.88 ± 0.13	2.86 ± 0.06	3.05 ± 0.05
	NEAA	37.40 ± 0.84	38.41 ± 1.75	36.82 ± 0.69	40.55 ± 0.48
	FAA	28.07 ± 0.61	27.44 ± 1.42	27.46 ± 0.61	28.90 ± 0.33
	TAA	35.01 ± 0.83	35.88 ± 1.76	34.35 ± 0.69	37.88 ± 0.43
	EAA/NEAA	1.07 ± 0.00 ^#^	0.99 ± 0.01	1.04 ± 0.013 ^#^	0.98 ± 0.00

^*^ Superscripted in means indicates significant differences between breeds (*p* < 0.05). ^#^ superscripted in means indicates significant differences between muscle parts within same breeds (*p* < 0.05). ^△^ Represents flavor amino acids. ^▲^ Represents tasty amino acids. HLG-BM, the breast muscle of Hy-Line grey chickens; HLG-TM, the thigh muscle of Hy-Line grey chickens; GYG-BM, the breast muscle of Guangyuan grey chickens; GYG-TM, the thigh muscle of Guangyuan grey chickens; Thr, threonine; Val, valine; Met, methionine; Ile, isoleucine; Leu, leucine; Phe, phenylalanine; Lys, lysine; His, histidine; Arg, arginine; Asp, aspartic; Ser, serine; Glu, glutamate; Gly, glycine; Ala, alanine; Cys, cystine; Tyr, tyrosine; Pro, proline. EAA, essential amino acid; NEAA, nonessential amino acid; FAA, flavor amino acids; and TAA, tasty amino acids.

## Data Availability

The data relevant to the study are available from the corresponding author upon reasonable request.

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
