# Peer review of "A Comparison of the Meat Quality, Nutritional Composition, Carcass Traits, and Fiber Characteristics of Different Muscular Tissues between Aged Indigenous Chickens and Commercial Laying Hens"

_foods, 2023, doi:10.3390/foods12193680_

Round 1

Reviewer 1 Report

The article titled "Comparison of Meat Quality, Nutritional Composition, Carcass Traits, and Fiber Characteristics of Different Muscular Tissues between Aged Indigenous Chickens and Commercial Laying Hens" presents an insightful investigation into the intrinsic differences between two distinct poultry groups—indigenous chickens and commercial laying hens. This comparative study encompasses multiple facets, ranging from meat quality and nutritional composition to carcass traits and fiber characteristics, thereby offering a comprehensive view of these poultry categories.

The article titled "Comparison of Meat Quality, Nutritional Composition, Carcass Traits, and Fiber Characteristics of Different Muscular Tissues between Aged Indigenous Chickens and Commercial Laying Hens" presents a commendable exploration of the differences between indigenous chickens and commercial laying hens. While the study covers an array of parameters and attributes, several improvements could elevate the rigor and overall quality of the research.

The introductory section establishes the relevance of the study but does contain non-technical terms such as "discharge." A careful review to ensure consistency with appropriate scientific terminology is advised. Furthermore, a crucial concern emerges regarding the sample size, as the number of birds might be insufficient to draw robust conclusions. Addressing this limitation would enhance the validity of the findings.

One of the most pressing issues pertains to the statistical analysis. The application of one-way ANOVA is questionable for this particular model. The authors should provide a thorough justification for selecting this analysis method or consider alternative statistical approaches more suitable for the study's design. This revision is essential to ensure the accuracy of the results and their interpretation.

The authors' inclusion of histological features is noteworthy but requires clarification. A more explicit rationale for incorporating this parameter is necessary, as it might not be directly relevant to the study's objectives. If deemed extraneous, it could be omitted to streamline the research focus and improve its coherence.

Another crucial aspect to address is the feed composition. Given the substantial impact of diet on meat quality, detailing the composition of the feeds offered to the poultry is imperative. By elucidating the dietary differences, the authors can establish a clearer link between nutritional intake and the observed variations in meat quality and other attributes.

Additionally, the authors' claim about the histological parameter needs further clarification. If the parameter is commonly performed in disease conditions and its relevance to this study is not well-established, it would be prudent to either justify its inclusion or consider omitting it from the analysis.

Reviewer 2 Report

The manuscript reports a study comparing two breed types of laying hens in terms of carcass and meat quality traits. This is against the background that old layers should represent a valuable source  of protein and other nutrients for human consumption such as they should be used. Therefore an array of parameters typically used to characterize carcass and meat quality is assessed.

Overall, the design is straight forward, e.g. comparison of a modern breed type vs. local breed type. Yet, a major weakness is the comparably low number of animals selected for detailed investigation.  Hence the power of the study appears to be low such as the relevance of the findings is considered rather low. This is regardless of the suggestion that ANOVA instead of t-testing should be applied (see comment below).

Material and methods need to be revised as quite some important details are missing.

·       Slaughter procedure and p.m. treatment should be described (as this affects pH development, color drip and tenderness)

·       Carcass traits: what is semi-eviscerated weight? (give the needed details so readers understand the procedure of dissection)

·       Colour measurement: give the necessary technical details (°observer, lighting D65…), blooming time etc.

·       Cooking loss: entire muscles cooked? Were the muscles packed in bags or simply cooked in water? => give the details

·       Tenderness measurement is missing relevant details including instrument settings, and how the samples were positioned.

·       Chemical analysis, e.g., HPLC determination of IMP is not properly described (eluents, machine settings, range and performance criteria of the calibration…)

·       Proximate composition: indicate whether results are presented related to fresh or dry matter

·       Histology: give the size of the fields analysed (or number of fibres counted)

Statistical analysis is not sound: as per the introduction, the study intended to compare breed types (then, t-test is fine). At the end, however, the study compared breed types and muscle types. Hence a two-way ANOVA (or GLM) including animal ID as random factor appears more appropriate. Suggest to redo the analysis (and accordingly modify display of results as well as modify discussion). In addition, next to significance effect sizes should be reported (e.g. Cohen’s d) such as the found differences can be judged in terms of their relevance instead of only stating whether significance was confirmed.

Conclusions are overstated. Need to be revised upon proper re-analysis of the data. Any statement regarding “better flavour” shall be deleted as no sensory evaluation or consumer testing was included. From IMP it only can be guessed that flavour is affected.

Some further detailed comments.

Line 33: is this statement relevant in the context of the study (EU statistics when presenting findings of local Chinese breeds)?

73: hens were selected for similar weight, so this isn’t really “random” anymore

115: freeze-dried?

Indicate the number of observations in all table headings and figure captions.

Fig. 2: the figure suggests that 2% IMP (related to fresh matter) were found in the meat . this appears implausibly high!

Finally, if all parameters were assessed for all muscle types/animals, a multivariate analysis (PCA) is recommended such as differences/similarities between breeds/muscle types as well as the underlying variables become easier to comprehend.

For the above reasons I suggest “MAJOR REVISION” before reconsidering the manuscript.

A moderate revision in terms of English usage is recommended

Reviewer 3 Report

-         The manuscript foods-2597494 entitled; “Comparison of Meat Quality, Nutritional Composition, Carcass Traits, and Fiber Characteristics of Different Muscular Tissues between Aged Indigenous Chickens and Commercial Laying Hens". The authors assess the differences in the meat quality, nutritional composition, carcass traits, and myofiber characteristics between Hy-Line grey (HLG, commercial breed) and Guangyuan grey (GYG, indigenous breed). The study is of interest in the field of meat quality. The study problem and methodology are generally well justified but, one of the main problems of this study is the small number of birds just 10 birds in every group???. The authors should revise their manuscript carefully regarding the following comments.  

General comments

-         The introduction section is too long, please rewrite it to be shorter. 

-         Please proofread the whole manuscript to avoid minor grammatical errors.

-         Please describe all abbreviations in their first mention.

-         The conclusion section is too long, please rewrite it to be shorter.

-         The discussion section is too long, please be more specific, discuss your study with other similar studies and please state the superiorities of your study when compared to previous ones.

Please proofread the whole manuscript to avoid minor grammatical errors.

Round 2

Reviewer 1 Report

The revised version is now acceptable for publication 

Author Response

我们已收到您的回复,非常感谢您对我们稿件的建议。

Reviewer 3 Report

All corrections are done as required. Thanks. 

Author Response

We have received your reply and thank you very much for your suggestions for our manuscript.